# Biodiversity-Driven Screening of Amphibian Skin Secretions for Inflammatory Modulation in Joint Diseases

**DOI:** 10.3390/toxins17090464

**Published:** 2025-09-17

**Authors:** Douglas Souza Oliveira, César Alexandre, Miryam Paola Alvarez-Flores, Isadora Maria Villas-Boas, Hugo Vigerelli, Isabel de Fátima Correia Batista, Michelle Cristiane Bufalo, Nancy Starobinas, Flávio Lichtenstein, Rafael Marques-Porto, Marcus Buri, Viviane Portas-Lopes, Pedro Luiz Mailho-Fontana, Marta Maria Antoniazzi, Denise Vilarinho Tambourgi, Ana Marisa Chudzinski-Tavassi, Catarina Teixeira, Carlos Jared, Olga Martinez Ibañez

**Affiliations:** 1Centre of Excellence in New Target Discovery (CENTD), Instituto Butantan, São Paulo 05503-900, Brazil; douglas.oliveira@butantan.gov.br (D.S.O.); miryam.flores@butantan.gov.br (M.P.A.-F.); isadora.boas@butantan.gov.br (I.M.V.-B.); hugo.barros@butantan.gov.br (H.V.); isabel.batista@butantan.gov.br (I.d.F.C.B.); michelle.bufalo@butantan.gov.br (M.C.B.); nancy.starobinas@butantan.gov.br (N.S.); flavio.lichtenstein@butantan.gov.br (F.L.); marcus.buri@butantan.gov.br (M.B.); denise.tambourgi@butantan.gov.br (D.V.T.); ana.chudzinski@butantan.gov.br (A.M.C.-T.); catarina.teixeira@butantan.gov.br (C.T.); 2Department of Biochemistry, São Paulo Federal University (UNIFESP), São Paulo 04023-900, Brazil; 3Laboratory of Structural Biology, Instituto Butantan, São Paulo 05503-900, Brazil; cesaralexandre01@hotmail.com (C.A.); pedro.fontana@fundacaobutantan.org.br (P.L.M.-F.); marta.antoniazzi@butantan.gov.br (M.M.A.); 4Laboratory of Immunochemistry, Instituto Butantan, São Paulo 05503-900, Brazil; 5Laboratory of Development and Innovation, Instituto Butantan, São Paulo 05503-900, Brazil; rafael.porto@butantan.gov.br (R.M.-P.); viviane.portas@fundacaobutantan.org.br (V.P.-L.); 6Historical Museum, Instituto Butantan, São Paulo 05503-900, Brazil; 7Laboratory of Immunogenetics, Instituto Butantan, São Paulo 05503-900, Brazil; 8Laboratory of Pharmacology, Instituto Butantan, São Paulo 05503-900, Brazil

**Keywords:** amphibian, inflammation, screening, chondrocytes, synoviocytes, macrophages

## Abstract

This study explores the direct effects of amphibian skin secretions on human cells involved in joint diseases, aiming to identify species with potential for inflammatory modulation. Secretions were obtained from sixteen species distributed across Brazilian biomes and one European species. Following biochemical characterization, human chondrocytes, synoviocytes, and macrophages were treated with secretions for 24 h. The cytotoxicity and modulation of the IL-6, IL-8, TNF-α, and IL-1β release were assessed. Synoviocytes showed the greatest resistance to cytotoxic effects, though sensitivity varied by species. Secretions from *Trachycephalus mesophaeus*, *Pipa carvalhoi*, and *Phyllomedusa bahiana* exhibited the highest cytotoxicity. At non-cytotoxic concentrations, *P. carvalhoi* and *Leptodactylus fuscus* strongly induced IL-6 and IL-8 in chondrocytes and synoviocytes, with *P. carvalhoi* also stimulating IL-1β and TNF-α release in macrophages. Among Bufonidae species, particularly *Rhinella jimi* and *Bufo bufo*, were potent inducers of TNF-α and IL-1β in macrophages. Secretions lacking pro-inflammatory effects were further tested for anti-inflammatory activity. *P. bahiana* reduced TNF-α production in stimulated macrophages and IL-6 in synoviocytes, while *Siphonops annulatus* and *T. mesophaeus* reduced LPS-induced TNF-α in macrophages. Our data underscore the rich biodiversity of amphibians, supporting the bioprospecting of their cutaneous secretions. These data reveal substantial potential for uncovering bioactive compounds with pharmacological applications.

## 1. Introduction

Currently, a total of 8863 amphibian species have been described around the world [1]. Brazil is a global leader in amphibian biodiversity, distributed in various regions and subjected to different habitat conditions. Amphibians known so far in the country comprise 1188 species, including 1144 species of anurans (frogs, toads, and tree frogs), followed by caecilians (39 species) and salamanders (5 species) [2]. These species represent an immensely rich source of bioactive compounds with technological potential.

Amphibian skin is a highly specialized organ with numerous granular (poison) and mucous glands that play key roles in respiration, water uptake, defense against pathogens and predators, and environmental adaptation [3]. Studies on the composition of these animals’ skin secretions revealed the presence of various chemical compounds, such as biogenic amines, alkaloids and steroids, peptides, and proteins [4,5]. These molecules have shown immunomodulatory, antimicrobial, anticancer, analgesic, and antidiabetic activities [6,7,8].

Inflammatory arthropathies encompass a wide range of joint disorders with a high global prevalence, and the most common forms are rheumatoid arthritis (RA) and osteoarthritis (OA). RA is the most common systemic autoimmune disease, affecting about 0.25–0.5% of the population, and OA is a complex degenerative disease, frequently compounded by the presence of multimorbidity [9,10]. In RA patients, the immune system attacks the joints. It creates a highly inflammatory environment that causes the joint tissues to thicken, resulting in swelling and pain in and around the joints, leading to permanent disability [11]. The precise pathogenesis of RA is unknown, but genetic and environmental factors are widely believed to play a role [12].

On the other hand, the complex pathogenesis of OA is well known and involves mechanical, inflammatory, and metabolic factors, which ultimately lead to structural destruction and failure [13]. During osteoarthritis, the cartilage composition changes, and the cartilage loses its integrity. The compositional changes alter the cartilage’s material properties and increase its susceptibility to disruption by physical forces. In an attempt to repair tissue loss, the hypertrophic chondrocytes exhibit increased synthetic activity but, in doing so, generate matrix degradation products and pro-inflammatory mediators that deregulate chondrocyte function and affect the adjacent synoviocytes, stimulating local proliferative and pro-inflammatory responses [14]. Proliferating synoviocytes, in turn, also release pro-inflammatory products, such as interleukin 6 (IL-6), a critical cytokine found to be elevated in the synovial fluid and serum of RA and OA patients. In OA, IL-6 is involved in the inflammatory processes in the synovium and cartilage, contributing to pain and the progression of the disease through its effects on cartilage turnover and inflammation [15,16]. Interleukin 8 (IL-8) is a chemotactic factor that mediates the inflammatory response, acting as a potent inducer of neutrophil infiltration into sites of inflammation. In both RA and OA, IL-8 is upregulated and is responsible for attracting neutrophils to the inflamed joint, where they release reactive oxygen species and proteases, further breaking down extracellular matrix (ECM) components and exacerbating tissue damage. This process is followed by tissue hypertrophy, increased vascularity, and bone tissue remodeling [17,18].

Macrophages are the most abundant immune cells within the synovial joints [19]. Synovial macrophages include resident and interstitial subsets, including the interstitial population of recruited monocyte-derived macrophages and the epithelial-like tissue-resident macrophages [20]. In this context, tumor necrosis factor-alpha (TNF-α) released mainly by macrophages is a central pro-inflammatory cytokine involved in the pathogenesis of both RA and OA, promoting the recruitment of immune cells into the synovium and stimulating the release of additional inflammatory mediators by synoviocytes and chondrocytes that perpetuate the inflammatory cycle [19,21]. Another key mediator produced by activated macrophages and other immune cells is Interleukin 1 beta (IL-1β), which plays a central role in driving inflammation in the joint. In RA, IL-1β induces the production of pro-inflammatory cytokines and enzymes, leading to synovial hyperplasia, cartilage degradation, and bone resorption. It is also implicated in pain in RA patients. In OA, IL-1β is involved in cartilage degradation by promoting the activity of MMPs and aggrecanases, which degrade ECM components [22]. Furthermore, IL-1β increases other inflammatory mediators’ production, sustaining the joint’s inflammatory environment [23,24].

The accumulation of these pro-inflammatory cytokines in the joint cavity is a key factor in the pathogenesis of inflammatory arthropathies. They activate various signaling pathways and pathological processes that generate a vicious cycle, ultimately leading to cartilage damage and pain by activating catabolic enzymes and pain mediators. Therefore, managing inflammation is always a relevant focus for therapy, and targeting TNF-α, IL-6, IL-8, and IL-1β remains a critical therapeutic strategy in managing RA and OA [25,26,27]. A range of drugs have been employed to treat chronic inflammatory joint diseases [6,28,29,30]. Although there have been significant advances in the treatment of arthritis over the past decade, many patients still fail to achieve sustained remission. While current therapies alleviate symptoms, they often cause severe side effects. There is a growing need for new and more effective drugs with fewer or no side effects.

To extend the research of new therapeutic targets for inflammatory joint diseases, in the present study, we analyzed the effect of skin secretions from 16 amphibian species from different regions of Brazil and one from Europe in cell cultures of human chondrocytes, synoviocytes, and macrophages. Skin secretions were added to the cell cultures at various concentrations to determine the cytotoxic potential. Non-toxic concentrations of the secretions were then used to evaluate their ability to modulate the synthesis of cytokines known to be involved in the pathogenesis of these inflammatory diseases.

## 2. Results

### 2.1. Secretions’ Protein Profile

Amphibian skin secretions contain different chemical compounds, and proteins are important players in the activities of animal poisons. Figure 1 shows the C18 reversed-phase chromatographic profile (monitored at 214 nm) (Figure 1A,C,E,G) and protein profiles of the different secretions separated by SDS-PAGE (Figure 1B,D,F,H). Using the same dry mass in the comparison, the figure illustrates the protein profiles and relative abundance between the secretions. The HPLC profiles show more than the SDS-PAGE, since they include peptides under 10 kDa and also some non-protein molecules. Some secretions, such as *Trachycephalus mesophaeus*, are highly proteic, while no protein bands are detectable in the Boana species, even in silver-stained gels. The protein molecular mass ranges from 14 kDa to more than 90 kDa and also varies considerably within the same family, though similarities are observed within the species, such as in *Rhinella* (*R. jimi*, *R. schneideri*, and *R. icterica*), *Boana* (*B. raniceps* and *B. albomarginata*), and *Dendropsophus* (*D. anceps* and *D. elegans*).

The available amounts of *Corythomantis greeningi* head and *C. greeningi* body (Family: Hylidae) secretions were small and not enough to continue the tests.

### 2.2. Cytotoxic Effect

To evaluate the toxicity of amphibian skin secretions on cells of interest, viability tests were carried out initially using different sample concentrations (2, 5, 10, and 25 or 50 μg/mL) after 24 h of incubation. Figure 2 shows the results of chondrocytes (Figure 2A), synoviocytes (Figure 2B), and macrophages (Figure 2C), respectively.

For chondrocytes, the highest concentrations of *Bufo bufo*, *Rhinella jimi*, *R. icterica*, *R. schneideri*, *Dendropsophus anceps*, and *D. elegans* skin secretions induced about 30 to 50% of the cell death, compared to the control. At these concentrations, the skin secretions of *Pipa carvalhoi*, *Phyllomedusa bahiana*, *Nyctimantis brunoi*, and *Trachycephalus mesophaeus* presented a high cytotoxic effect, inducing death of about 90% of the cells.

In synoviocytes, *Phyllomedusa bahiana* and *Nyctimantis brunoi*, at the highest concentration, induced death in about 50% of the cells, and only *Trachycephalus mesophaeus* presented high toxic activity, causing 80% of the cell death at all concentrations tested compared to the control.

For macrophages, *Pristimantis paulodutrai*, *Phyllomedusa bahiana*, *Bufo bufo*, *Rhinella icterica*, *R. schneideri*, *Dendropsophus elegans*, *Nyctimantis brunoi*, and *Trachycephalus mesophaeus* skin secretions were toxic at the highest concentration, inducing the death of about 75 to 90% of the cells compared to the basal control.

*Trachycephalus mesophaeus*, *Phyllomedusa bahiana*, and *Nyctimantis brunoi* skin secretions were cytotoxic to the three cell types at the concentrations tested. Interestingly, some secretions such as *Siphonops annulatus*, *Leptodactylus fuscus*, *Boana albomarginata*, *B. raniceps*, and *Itapotihyla langsdorfii* were not toxic to any of the cell types at the highest dose of 25 μg/mL. For the other secretions, there was variation between the cells, with synoviocytes being the most resistant to the toxic effects of most secretions. Interestingly, secretions of *Pristimantis paulodutrai* and *Bufo bufo* presented cytotoxic effects only in macrophages and *Pipa carvalhoi* only in chondrocytes. The results are summarized in the heatmap (Figure 2D).

Additional viability assays were performed with those secretions that presented cytotoxic activity at the initial concentrations tested to find non-cytotoxic (>85% viability) doses to be tested for subsequent cytokine assays, Appendix A.

### 2.3. Cytokine Release

The release of IL-6 and IL-8 was evaluated after the stimulation with secretions to characterize the amphibian skin secretion’s pro-inflammatory effects in chondrocytes and synoviocytes. Among the 16 species tested, *Pipa carvalhoi*, *Leptodactylus fuscus*, and *Dendropsophus anceps* induced cytokine production in chondrocytes (Figure 3). *P. carvalhoi* at 10 and 25 μg/mL increased the release of IL-6 by 140- to 170-fold and of IL-8 at 25 μg/mL by 420-fold. *L. fuscus* also increased IL-6 production at 10 and 25 μg/mL by 70- to 80-fold. Moreover, the secretion from *D. anceps* increased the IL-8 release by 490-fold. *P. carvalhoi* and *L. fuscus* also induced a significant IL-6 release at its highest concentration in synoviocytes by 20- to 34-fold. An increase in IL-8 production was observed after *P. carvalhoi* and *L. fuscus* stimuli (Figure 4).

The levels of TNF-α, IL-6, and IL-1β in supernatants of macrophage cultures induced by non-cytotoxic doses of skin secretions are presented in Figure 5. *Bufo bufo* was the only secretion that significantly increased the TNF-α release at 0.02 and 0.05 μg/mL (around 30-fold), compared to the control group. The skin secretions from *Pipa carvalhoi* (at 10 and 25 μg/mL) and from Bufonidae species, such as *B. bufo* (at 0.02, 0.05, and 0.1 μg/mL), *Rhinella jimi* (at 5, 10, and 25 μg/mL), *R. icterica* (at 0.1, 0.2, and 0.5 μg/mL), and *R. schneideri* (at 0.2, 0.5, and 1 μg/mL), significantly increased the IL-1β release (50 to 200-fold). Interestingly, Rhinella species induced a potent IL-1β release compared to the other skin secretions. These secretions from Bufonidae species seem to induce IL-6 release.

### 2.4. Anti-Inflammatory Effect of Skin Secretions

Among the skin secretions that did not induce considerable cytokine production in the previous tests, some were elected to explore the anti-inflammatory activity. Samples from *Siphonops annulatus*, *Pristimantis paulodutrai*, *Phyllomedusa bahiana*, *Itapotihyla langsdorfii*, and *Trachycephalus mesophaeus* were added to the cells 1 h after the stimulus with IL-1β, for chondrocytes (Figure 6) and synoviocytes (Figure 7), or LPS for macrophages (Figure 8). Dexamethasone was used as a positive control for pro-inflammatory cytokine inhibition. Of all these, the *P. bahiana* skin secretion at 0.05 μg/mL reduced the IL-6 (96%) and IL-8 (51%) release by IL-1β-stimulated synoviocytes and reduced the TNF-α (82%) release by LPS-stimulated macrophages.

*Siphonops annulatus* (25 μg/mL) and *Trachycephalus mesophaeus* secretions (0.05 μg/mL) also significantly decreased, by about 65 and 25%, respectively, the TNF-α release by LPS-stimulated macrophages. Interestingly, the secretion of *Pristimantis paulodutrai* at 10 μg/mL maximizes the LPS-induced TNF-α and IL-6 release. *Phyllomedusa bahiana* and *T. mesophaeus* secretions also increased the IL-6 production in macrophages at the lowest concentration tested. Regarding the macrophage LPS-induced IL-1β release, *P. bahiana* and *T. mesophaeus* secretions induced an increased interleukin release at the three concentrations tested: *P. bahiana* at 0.02, 0.05, and 0.1 μg/mL and *T. mesophaeus* at 0.1, 0.2, and 0.5 μg/mL.

## 3. Discussion

Amphibian skin secretions represent a rich reservoir of bioactive compounds, prompting extensive research into their potential as therapeutic agents or drug leads. This study examined the immunomodulatory effects of skin secretions from 14 species from Order Anura and 1 from Order Gymnophiona (caecilians). They were collected across diverse Brazilian biomes (Neotropical fauna) and alongside the European (Paleoarctic fauna) *Bufo bufo*. This inclusion aimed to identify potential variations in the compound composition or bioactivity. Notably, *Siphonops annulatus*, a caecilian, was included to broaden the scope beyond anurans. The present work aligns with the growing global interest in bioprospecting biodiversity for potential pharmacological applications. We analyzed the protein profiles and the cytotoxic activities, and we assessed the capacity of these secretions to modulate the pro-inflammatory cytokine production in chondrocytes, synoviocytes, and macrophages.

The electrophoretic analysis revealed a diverse protein profile across the secretions, with molecular masses ranging from 14 kDa to over 90 kDa (Figure 1). While the protein composition varied interspecifically, similarities among species of the same genus were observed in *Rhinella*, *Boana*, and *Dendropsophus*. As expected, due to the evolutionary distance among orders in the class Amphibia, the caecilian *Siphonops annulatus* exhibited a distinct protein composition compared to the anuran species. While the electrophoretic profiles reveal only proteins above 10 kDa, the HPLC profiles show a better picture of the molecular complexity of these secretions. Still the secretions from species closer to one another are, as expected, more similar than to more distant ones, and those with fewer bands in SDS-PAGE are also less complex in the HPLC chromatograms. *Phyllomedusa bahiana* stands out with the most complex HPLC profile, in contrast with its SDS-PAGE profile, which is quite simple. This is probably due to the fact that its content is mostly peptides and non-protein molecules [31,32,33]. In zoological terms, this comparative information can also contribute to the understanding and study of amphibian physiology and immunology. These profiles suggest the potential presence of similar compounds in secretions that not only share biochemical characteristics but also exhibit comparable effects on cultured cells.

Subsequent cytotoxicity assays demonstrated differential sensitivity among cell lines tested, with synoviocytes displaying the highest resistance (Figure 2). *Trachycephalus mesophaeus*, *Pipa carvalhoi*, and *Phyllomedusa bahiana* exhibited significant cytotoxicity across all cell lines, with *T. mesophaeus* displaying the most cytotoxic potential. Although the composition of *T. mesophaeus* secretions remains largely uncharacterized, the presence of hyaluronidase was described in the Lophyohylini species that is exclusive among anurans [34]. Studies on *T. venulosus* have documented the toxic effects of its skin secretions on human eyes [35]. Moreover, studies with *P. carvalhoi* secretions characterized kynurenic acid, a tryptophan metabolite, as the primary content, and the absence of peptides was described [36]. High concentrations or prolonged exposure to kynurenic acid have been associated with neuronal cellular damage [37,38]. On the other hand, the *Phyllomedusa* genus exhibits remarkably complex skin secretions, containing the highest diversity and abundance of bioactive peptides found in any amphibian. These secretions are peptide-rich and contain relatively few proteins with a broad spectrum of biological activities, including cytotoxic, antimicrobial, antiviral, antitumoral, and antinociceptive effects [32,39,40,41].

To examine the pro-inflammatory potential of amphibian skin secretions, we tested non-cytotoxic doses to stimulate chondrocytes, synoviocytes, and macrophages, quantifying cytokine release (Figure 3, Figure 4 and Figure 5). Readouts for chondrocytes and synoviocytes were IL-6 and IL-8, and TNF-α, IL-1β, and IL-6 were evaluated in macrophages. These cytokines play pivotal roles in the pathogenesis of chronic inflammatory joint diseases. In such conditions, monocyte-derived macrophages, prevalent within inflamed joint tissues, contribute significantly to the inflammatory response, producing high amounts of TNF-α and IL-1β, signaling synoviocytes and other cells in the joint tissue. Concurrently, synoviocytes release pro-inflammatory mediators, including prostaglandins, nitric oxide, and cytokines, exacerbating joint swelling, pain, and fibrosis [42]. Furthermore, during the progression of articular inflammation, chondrocytes undergo morphological and functional changes, becoming hypertrophic and producing type I collagen and higher levels of cartilage-degrading enzymes than normal chondrocytes. The digested matrix protein fragments are released into the synovial fluid and stimulate macrophages, synovial fibroblasts, and chondrocytes to produce inflammatory mediators, feeding this vicious, inflammatory cycle and leading to joint pain, stiffness, and cartilage destruction [43].

Regarding pro-inflammatory activities, the skin secretions of the *Pipa carvalhoi* and Leptodactylidae family exhibited the highest activity in chondrocytes and synoviocytes, triggering the release of IL-6 and IL-8. (Figure 3 and Figure 4) *P. carvalhoi* is an anuran species from the Pipidae family, endemic to Brazil, specifically from the Atlantic Rainforest. Adapted to aquatic environments, these frogs fertilize and incubate their eggs on the backs of the females, where the embryos develop until they hatch [44]. While little is known about the composition of *P. carvalhoi* skin secretions, our study on specimens collected in the State of Bahia revealed that most components are concentrated in the low-molecular-weight range. A previous study using liquid chromatography and mass spectrometry identified several components below 800 Da, revealing that *P. carvalhoi* does not produce peptides as toxins in its skin [45]. The primary component in its secretion is kynurenic acid, which has been suggested to regulate arthritic and other inflammatory processes [46]. However, in our study, the *P. carvalhoi* skin secretion also stimulated the IL-1β release from macrophages (Figure 5). The pro-inflammatory effects evidenced in our analysis of the whole skin secretion may result from the prevalence of activity of another component present in this secretion. On the other hand, the Leptodactylidae family is one of the most diverse frog families in the Neotropics. Their skin secretions contain various substances, including amines, such as histamine and tryptamine derivatives, neuropeptides, and antimicrobial peptides. Some components have shown activity against pathogenic microorganisms like *Escherichia coli* and *Staphylococcus aureus*, while others exhibit immunomodulatory and antioxidant effects [47]. Despite the limited knowledge on these substances’ functions and evolutionary significance, our study of specimens collected in Bahia revealed that the Leptodactylidae skin secretion induces the IL-1β release by macrophages (Figure 5).

In macrophages, Bufonidae species, particularly *Bufo bufo* and *Rhinella* species, exhibited the most potent pro-inflammatory activity, inducing TNF-α and IL-1β release. *R. jimi* displayed the highest potency in IL-1β induction (Figure 5). Bufonidae secretions, rich in bufadienolides, alkaloids, and biogenic amines, have demonstrated diverse pharmacological properties, including antimicrobial, anticancer, and anti-inflammatory effects [48,49,50,51,52,53]. Components of the *R. schneideri* poison activated the complement cascade and generated the SC5b-9 membrane attack complex [54]. The secretions of *R. schneideri* collected in the State of São Paulo tested in our study were highly toxic to macrophages and chondrocytes (Figure 2) and very effective at inducing the release of cytokines by macrophages at low doses (Figure 5).

To investigate potential anti-inflammatory activities, secretions that did not induce pro-inflammatory effects were assessed for their ability to suppress the cytokine release induced by IL-1β or LPS. *Phyllomedusa bahiana* reduced the TNF-α release in LPS-stimulated macrophages (Figure 8) and the IL-6 release in IL-1β-stimulated synoviocytes (Figure 7). *Siphonops annulatus* and *Trachycephalus mesophaeus* also reduced the LPS-induced TNF-α release in macrophages (Figure 8). Several *Phyllomedusa* species have been widely studied due to the high content of bioactive substances stored in their skin glands. In popular belief, *P. bicolor* secretions are used in the Kambo or Sapo shamanic rituals of Amazonian Indigenous peoples, introducing the poison through burns in the skin [55]. Scientifically, peptides with antimicrobial and antiprotozoal actions have been described in secretions from this species [56]. Here, using *P. bahiana*, collected in the State of Bahia, we evidenced a likely anti-inflammatory activity on the cytokine release by macrophages and synoviocytes. Examples of peptides derived from *Phyllomedusa* skin exhibiting such activities include dermaseptins, dermatoxins, phylloseptins, and phylloxins. Furthermore, *Phyllomedusa* skin secretions contain peptides structurally, though not necessarily biosynthetically, related to mammalian bradykinins, tachykinins, cholecystokinin, corticotropin-releasing hormone, and opioid peptides, demonstrating antinociceptive potential [57]. Authors found immunomodulatory effects in Phylloseptin and plasticin, two synthetic peptides first isolated from *P. trinitatis*. These molecules were demonstrated to modulate the production of pro-inflammatory cytokines, decreasing TNF-α and IL-1β release and increasing anti-inflammatory cytokine interleukin-10 (IL-10) by mouse peritoneal cells, lyse human tumor-derived cells, and mouse erythrocytes [58]. The class of peptides called plasticins was first identified in skin secretions of the phyllomedusin frogs *Agalychnis callidryas* and *P. bicolor*. Still, subsequently, plasticin-L1 was isolated from skin secretions of the South American Santa Fe frog *Leptodactylus laticeps* [58]. Those plasticins with a positive charge at the physiological pH show a broad-spectrum antimicrobial activity, and plasticin-L1 increased the production of the pro-inflammatory cytokines TNF-α, IL-1β, IL-12, and IL-23 by mouse peritoneal macrophages [59]. In our study, the *P. bahiana* skin secretion also potentiated the LPS-induced IL-1β release in macrophages. The presence of peptides with both pro- and anti-inflammatory activities may explain these complex effects. Similar results were obtained with skin secretions from *T. mesophaeus* (milk tree frog) (Figure 8). They are endemic to Brazil, inhabiting low-lying areas of tropical and subtropical rainforests. Skin glands secrete a milky, volatile, noxious, and alkaline skin secretion when handled, irritating mucous membranes [60]. These anurans are sticky and use this natural “glue” as a defense mechanism. In our assays, the skin secretion from specimens collected in the State of Bahia reduced the release of IL-6 and TNFα at the highest concentration of 0.2 μg/mL and increased the IL-1β release by stimulated macrophages.

*Siphonops annulatus* secretions exhibited a dose-dependent reduction in the LPS-induced TNF-α release in macrophages (Figure 8). From an adaptive point of view, the animal lives in tunnels in an underground environment. It is presumable that they naturally have anti-inflammatory components since they are constantly rubbing their skin against the walls of the tunnels. This same fossorial behavior, living in an environment conducive to the proliferation of microorganisms, makes it possible for them to also be producers of antimicrobial substances. This skin is moist and can be compared to a culture medium [61]. Interestingly, despite originating from different taxonomic orders (Anura and Gymnophyona) and being phylogenetically very distant, the skin secretions exhibited comparable effects in reducing the TNF production by activated macrophages. This similarity may result from the action and/or synergistic interaction of distinct bioactive compounds present in these secretions.

Of note, the skin secretions of *Phyllomedusa bahiana* and *Trachycephalus mesophaeus* reduced the TNF-α production but increased IL-1β production (Figure 8B–D). Specific stimuli (poison component) could preferentially activate signaling pathways, leading to IL-1β production while inhibiting those leading to TNF-α. Protein degradation, or transcriptional regulation as a function of time, could also influence this balance between cytokines, in addition to the complexity of the composition of the secretions, which may contain substances with pro- and anti-inflammatory effects.

## 4. Conclusions

This study presents an unprecedented exploratory analysis of amphibian skin secretions from a diverse range of species, including several that are increasingly threatened by ecological changes. The potential loss of these species underscores the urgency of bioprospecting their rich chemical diversity. Our findings demonstrate that these secretions contain bioactive components capable of directly modulating inflammatory responses in human macrophages, synoviocytes, and chondrocytes, cell types critically involved in joint diseases such as osteoarthritis and rheumatoid arthritis. Of note, *Pipa carvalhoi* and species from the families Leptodactylidae and Bufonidae exhibited a significant pro-inflammatory effect by inducing cytokine release. In contrast, others, including *Siphonops annulatus*, *Phyllomedusa bahiana*, and *Trachycephalus. mesophaeus*, induced a decrease in the cytokine release by activated cells, suggesting a likely anti-inflammatory effect. Interestingly, the latter two skin secretions decreased the TNF-α and IL-6 production while simultaneously increasing IL-1β synthesis, indicating the presence of both pro- and anti-inflammatory substances within the secretions. The diversity of species and biological activities observed provides a valuable platform for identifying novel therapeutic targets and signaling pathways. These insights may contribute to the development of innovative strategies for managing inflammatory joint disorders.

## 5. Materials and Methods

### 5.1. Amphibian Skin Secretions

Previous authorizations for animal collection and skin secretions were provided by the Instituto Chico Mendes de Conservação da Biodiversidade SISBIO (36375-7, 15964-6, 65237-7, 48080-4) and SISGEN A16AA7A. Fieldwork involved traveling over 4000 km across Brazil to access remote and poorly explored habitats, where many species remain virtually unknown.

All details about animals, secretion collection, and voucher numbers are summarized in the Appendix A. The number of specimens of each species used for skin secretion extraction varied from 3 to 6 specimens, depending on availability. All specimens were adults and did not have their sex determined. The specimens were kept in the Vivarium of the Structural Biology Laboratory at the Butantan Institute in 45 L plastic boxes enriched with dry branches as perches (in the case of tree frogs), plastic or rubber tubes for shelter, and water that was replenished daily. The animals were fed once a week with varying amounts of crickets (*Gryllus assimilis*), cockroaches (*Nauphoeta cinerea*), and newborn mice, depending on the species. The secretions of *Pristimantis paulodutrai*, *Dendropsophus anceps*, and *D. elegans* were obtained during fieldwork immediately after capture of the specimen, followed by its release. Specimens (n = 2–3) of each *Dendropsophus* species were euthanized and deposited in the zoological collection of the Laboratory of Structural Biology as vouchers. The secretions of the other species were obtained from the animals maintained in captivity in the same laboratory.

The lyophilized skin secretions from *Siphonops annulatus*, Pipa carvalhoi, *Leptodactylus fustus*, *Pristimantis paulodutrai*, *Bufo bufo*, *Rhinella schneideri* (from savannah, currently *R. diptycha*), *Rhinella jimi* (from semiarid, currently *R. diptycha*), *Rhinella icterica*, *Phyllomedusa bahiana*, *Boana raniceps*, *Boana albomarginata*, Dendropsophus anceps, *Dendropsophus elegans*, *Corythomantis greeningi* (CG body; CGh head), *Itapotihyla langsdorfii*, *Nyctimantis brunoi* (Nb), and *Trachycephalus mesophaeus* were provided by the Structural Biology Laboratory from Butantan Institute. Briefly, the crude skin secretion was obtained by gentle manual stimulation of the whole body of the animals, one at a time, for 5 min inside a plastic bag containing 50 mL of deionized water, forming a pool that was next frozen at −20 °C, stored at −80 °C, and lyophilized prior to the experiments. For *R. jimi*, *R. shneideri*, and *R. icterica*, the secretion was obtained exclusively from the parotoid macroglands by manual compression and placed directly into plastic tubes. The lyophilized secretion of Bufo bufo was purchased from Latoxan Laboratories S.A.S. (Portes lès Valence, France, product # ID L3101). The samples were resuspended in apyrogenic saline solution, and protein content was determined by absorbance using a nanodrop spectrophotometer. All amphibian secretions tested were lipopolysaccharide (LPS)-free. The LPS contents of the secretion samples (5.0 µg/mL) were assayed in the “In vitro Laboratory of the Instituto Butantan Quality Control Service” using the Gel-Clot Assay Kit.

### 5.2. SDS-PAGE and HPLC Analyses

All studied samples were submitted to Reversed-Phase High-Performance Liquid Chromatography (RP-HPLC) and SDS-PAGE analysis to evaluate the peptide and protein profiles of the crude poisons. To obtain the RP-HPLC profiles, 50 μL of each secretion (1 mg/mL) was loaded into a Phenomenex C18 (250 × 4.6 mm) column, equilibrated with 0.1% TFA (solvent A), and eluted with a 0–100% gradient of 90% ACN/0.1% TFA (solvent B), monitored at 214 nm. The skin secretions (10 μg, dry weight) were analyzed by 12% SDS-PAGE under reducing conditions (2% β-mercaptoethanol). The gels were either Coomassie Brilliant Blue- or silver-stained.

To characterize the peptide and protein composition of the crude skin secretions, all samples were analyzed by Reversed-Phase High-Performance Liquid Chromatography (RP-HPLC) and SDS-PAGE. For RP-HPLC profiling, 50 μL aliquots of each secretion (1 mg/mL) was injected onto a Phenomenex C18 column (250 × 4.6 mm), previously equilibrated with 0.1% trifluoroacetic acid (TFA—solvent A) and eluted using a linear gradient from 0 to 100% of solvent B (90% acetonitrile containing 0.1% TFA), monitored at 214 nm. For SDS-PAGE analysis, 10 μg (dry weight) of each skin secretion was loaded per lane in 12% gels under reducing conditions using 2% β-mercaptoethanol. Gels were stained using either Coomassie Brilliant Blue or silver staining to enhance protein visualization.

### 5.3. Cell Culture

For the screening assays using amphibian skin secretions, three human cell models were used: primary articular synoviocytes, primary chondrocytes, and the THP-1 cell line differentiated into macrophages.

#### 5.3.1. Chondrocytes

Primary normal human articular chondrocytes (NHAC-Kn, #CC-2550, Lot 8F3336) were purchased from Lonza (Walkersville, MD, USA) at passage 2. Cell expansion and subculturing were performed following the manufacturer’s guidelines, utilizing CBM Chondrocyte Growth Basal Medium (#CC-3217, Lonza, Basel, Switzerland) supplemented with CGM Single-Quots (#CC-3217). Cells were cryopreserved at passage 5. For experimental assays, chon-drocytes at passage 6 were plated into 96-well plates (Costar^®^, Corning Inc., Tewksbury, MA, USA) at a density of 1 × 10^4^ cells/well and maintained in Dulbecco’s Modified Eagle Medium/Nutrient F-12 Ham (DMEM/F12, Gibco #11320033, Bleiswijk, The Netherlands), supplemented with 5 mM HEPES and 10% fetal bovine serum (FBS, #12657029, Gibco, Grand Island, NY, USA), and incubated at 37 °C with 5% CO_2_ in a humidified atmosphere for 24 h. Then, the culture medium was replaced with a starvation medium (DMEM/F12 containing 1% FBS), and chondrocytes were submitted to the treatment according to screening assays described below.

#### 5.3.2. Synoviocytes

Human B-type synoviocytes (CDD-H-2910-N, cryopreserved normal donor cells) were purchased from Articular Engineering (Chicago, IL, USA). Cells were expanded and subcultured in Dulbecco’s Modified Eagle’s Medium/Nutrient F-12 Ham (DMEM/F12, Sigma) supplemented with 10% fetal bovine serum (FBS; Cultilab, Campinas, SP, Brazil). At passage 6, synoviocytes were plated into 96-well plates (Costar^®^, Corning Inc., Tewksbury, MA, USA) at a density of 1 × 10^4^ cells/well and incubated in a humidified atmosphere at 37 °C with 5% CO_2_ for 24 h. Subsequently, the culture medium was replaced with a starvation medium consisting of DMEM/F12 supplemented with 1% FBS, and cells were treated according to the screening assay.

#### 5.3.3. THP-1 Macrophages

The THP-1 human monocytic cell line was acquired from the American Type Culture Collection (ATCC, Manassas, VA, USA). Cells were maintained in RPMI 1640 medium (Sigma-Aldrich, #R6504), supplemented with 10% fetal bovine serum (FBS; Gibco, Grand Island, NY, USA #26140079), 2 mM L-glutamine (Sigma-Aldrich, St. Louis, MO, USA #A2916801), and 1 mM sodium pyruvate (Sigma-Aldrich, St. Louis, MO, USA #P5280), and cultured at 37 °C in a humidified incubator with 5% CO_2_.

To induce macrophage differentiation, THP-1 cells were incubated with 50 nM phorbol 12-myristate 13-acetate (PMA; Sigma-Aldrich, St. Louis, MO, USA #P8139). For this, 2 × 10^4^ cells were seeded into 96-well plates (Costar^®^, Corning Inc., Tewksbury, MA, USA) in PMA-containing medium and incubated for 48 h. Following differentiation, the medium was replaced with PMA-free medium, and the adherent macrophage-like cells were rested for an additional 24 h prior to treatment with amphibian skin secretions.

### 5.4. Screening Assays

Two types of assays were carried out on each cell, one to evaluate the ability of secretions to induce pro-inflammatory cytokine production and another assay to evaluate possible anti-inflammatory effect by measuring cytokine production inhibition after previous pro-inflammatory stimulus. To evaluate pro-inflammatory effects in chondrocyte and synoviocyte cultures, cells were treated with skin secretions in three different concentrations and incubated for 24 h. IL-1β at 1 ng/mL was used as positive control. Furthermore, for the anti-inflammatory effect assay, chondrocytes and synoviocytes were incubated 1 h before with IL-1β at 1 ng/mL. Cells were treated with amphibians’ skin secretions at three different concentrations and incubated for 24 h. Dexamethasone at 1 µM was used as anti-inflammatory positive control.

In the case of THP-1 differentiated macrophages, cells were treated with secretions at three different non-cytotoxic concentrations for 24 h. LPS at 10 ng/mL was used as a pro-inflammatory positive control. For the anti-inflammatory effect assay, macrophages were incubated with LPS at 10 ng/mL 1 h before then treated with amphibian skin secretions and incubated for 24 h. Dexamethasone at 1 µM was used as anti-inflammatory positive control. After the incubation, supernatants from all cell-type cultures were collected for cytokine production analysis, and cells were used for viability tests. TNF-α, IL-6, and IL-1β releases were evaluated in macrophage supernatants, while IL-6 and IL-8 production was evaluated for chondrocytes and synoviocytes as described below. Most assays were conducted in triplicate across two independent experiments to ensure reproducibility.

### 5.5. Cell Viability

For cell viability analysis, supernatants from cell cultures incubated with amphibian skin secretions and cells treated with saline (basal control) were removed and incubated with 3-(4.5-Dimethylthiazol-2-yl)-2,5-diphenyl tetrazolium bromide assay (MTT, 0.5 mg/mL in culture medium, Sigma-Aldrich, St. Louis, MO, USA) for 3 h at 37 °C and 5% CO_2_ [62]. This assay quantifies the levels of mitochondrial activity as the readout of cell viability. The formazan blue crystals were solubilized with 100 µL dimethylsulfoxide (DMSO, Merck, Darmstadt, Germany) for 10 min, and absorbance was quantified at 540 nm using SpectraMax microplate reader (Molecular Devices, LLC. San Jose, CA 95134). The percentage of cell viability of secretion-treated cells was extrapolated from saline-treated cells (basal control = 100% viability). MTT assays were repeated until a non-toxic dose (cell viability > 85%) was identified for each secretion.

### 5.6. Evaluation of Cytokines Release

Concentrations of IL-1β, IL-6, IL-8, and TNF-α cytokines were determined in cell-free supernatants from chondrocyte, synoviocyte, and THP-1 macrophage cell cultures by bead-based multiplex analysis using the Milliplex MAP Human Cytokine/Chemokine Magnetic Bead Panel, Catalog #HCYTOMAG-60K (Millipore), according to the manufacturer’s instructions. The Luminex xPONENT 4.3 software was used for data acquisition and the MILLIPLEX Analyst 5.1 for data analysis. The concentrations of cytokines (pg/mL) were interpolated from standard curves, Appendix A. Relative cytokine concentrations are reported as fold change means (FC). In the pro-inflammatory experiments, FC is calculated by dividing the value of cytokine concentration in pg/mL obtained for each sample by the mean value of the negative control (cells treated with PBS). For the anti-inflammatory experiments, the cytokine concentration for each sample is divided by the mean value of the positive control. The positive control consists of cells stimulated with IL-1β for chondrocytes and synoviocytes and LPS for macrophages.

### 5.7. Statistical Analysis

Cell viability data are expressed as mean ± confidence interval error from the replicates of two independent experiments. We applied two-tailed Student’s *t*-test with an 80% confidence level to estimate the errors of the bars. For data visualization, heatmaps were generated in RStudio 2025.05 version using the heatmap package, with a color scale ranging from 20% (red) to 120% (green) viability relative to control.

Relative cytokine concentrations, using non-toxic doses of amphibian secretions, are presented as the mean fold change from the replicates of two independent experiments, with error bars representing the confidence interval. We applied two-tailed Student’s *t*-test with an 80% confidence level to estimate the errors of the bars. Since many sample sizes were below 5, we used the non-parametric Kruskal–Wallis test followed by Dunn’s test, without correction for multiple comparisons, to provide a semi-quantitative assessment of the screening. The following *p*-values indicate statistical significance: * for 0.20 > *p* ≥ 0.05, ** for 0.05 > *p* ≥ 0.01, *** for 0.01 > *p* ≥ 0.001, and **** for *p* < 0.001. All statistical calculus was performed using Python 3.11.11, SciPy 1.15.1, and Scikit-post hocs 0.11.2.

## Figures and Tables

**Figure 1 toxins-17-00464-f001:**
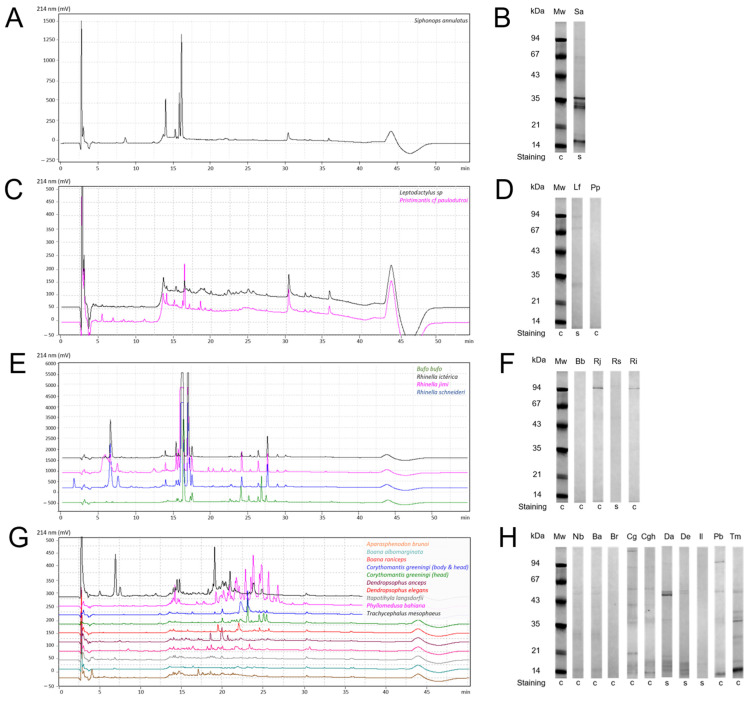
BRP-HPLC comparisons of the chromatographic profiles of the studied amphibians’ skin secretions from orders Gymnophiona and Anura; (**A**) Siphonopidae, (**C**) Leptodactylidae, (**E**) Bufonidae, and (**G**) Hylidae. The skin secretions were analyzed by RP-HPLC on a C18 column using a 0 to 90% acetonitrile gradient in 0.1% trifluoroacetic acid, on a Shimadzu Prominence 20A binary system. The profiles were processed, analyzed, and compared using Shimadzu’s proprietary software, LabSolutions. Version 5.86. SDS-PAGE analysis of secretions (**B**,**D**,**F**,**H**). Secretions from *Siphonops annulatus* (Sa), *Pipa carvalhoi* (Pc), *Leptodactylus fuscus* (Lf), *Pristimantis paulodutrai* (Pp), *Bufo bufo* (Bb), *Rhinella jimi* (Rj), *R. schneideri* (Rs), *R. icterica* (Ri), *Phyllomedusa bahiana* (Pb), *Boana raniceps* (Br), *B. albomarginata* (Ba), *Dendropsophus anceps* (Da), *D. elegans* (De), *Corythomantis greeningi* (Cg body; Cgh head), *Itapotihyla langsdorfii* (Il), *Nyctimantis brunoi* (Nb), and *Trachycephalus mesophaeus* (Tm) (50 μL, 1 mg/mL) were run in 12% polyacrylamide gels. After the run, the gel was either silver (s)- or Coomassie blue (c)-stained, as indicated below the strips. Molecular weight standard (MW) in kDa. The uncropped gel image is available in Appendix A.

**Figure 2 toxins-17-00464-f002:**
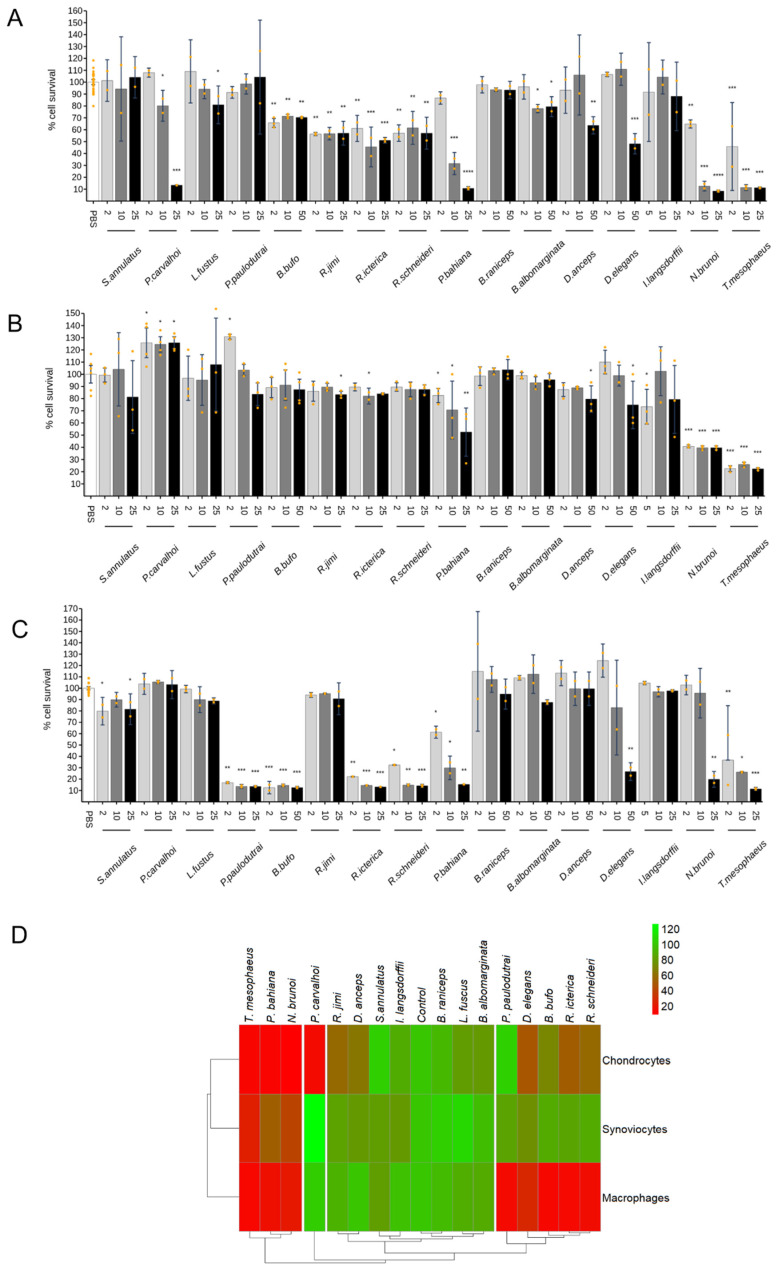
Effect of amphibian’s secretion on chondrocytes’ (**A**), synoviocytes’ (**B**), and macrophages’ (**C**) viability. Cells were treated with amphibian secretions at different concentrations for 24 h. The MTT assay measured cellular viability. Each data point is presented as the mean ± confidence interval error from two independent experiments. Statistical analysis was performed using Dunn’s test without correction for multiple comparisons. The *p*-values are indicated as follows: * for 0.2 > *p* > 0.05, ** for 0.05 > *p* ≥ 0.01, *** for 0.01 > *p* ≥ 0.001, and **** for *p* < 0.001. The heatmap summarizes the cytotoxic effects (**D**) of the highest concentration of each amphibian skin secretion tested on macrophages, synoviocytes, and chondrocytes. The color scale represents cell viability compared to control, ranging from 20% (**red**) to 120% (**green**). Appendix A presents the cell death control using staurosporine.

**Figure 3 toxins-17-00464-f003:**
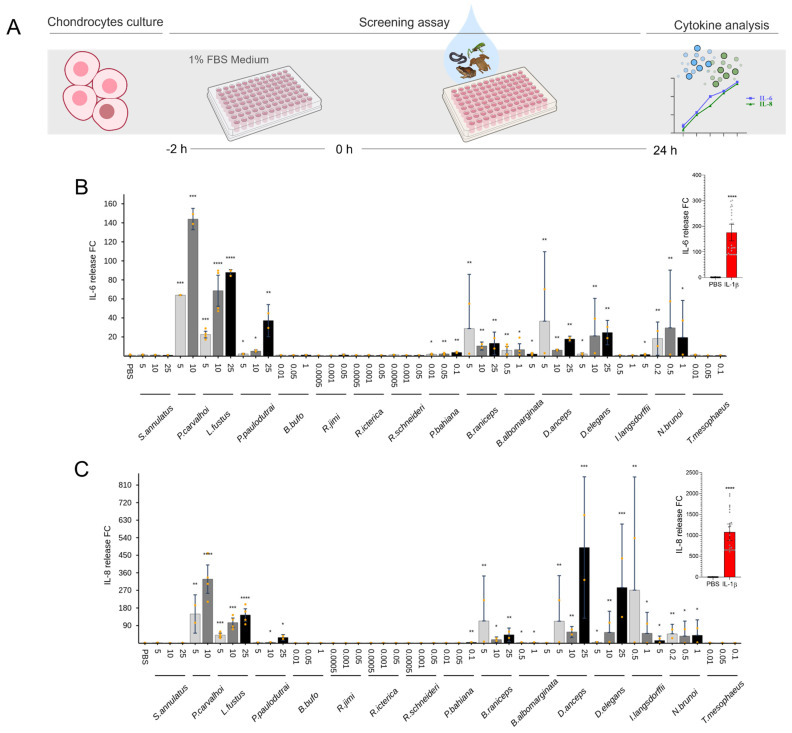
Effect of amphibian secretions on release of IL-6 and IL-8 by chondrocytes. Culture supernatants were collected 24 h after incubation with each secretion to assess the content of cytokines. (**A**) Assay workflow. (**B**) IL-6. (**C**) IL-8. IL-1β was used as positive control, and results are represented in the small graphs on the right. Each data point is presented as the mean fold change (FC) from the replicates of two independent experiments relative to cytokine levels of controls (cells in PBS), with error bars representing the confidence interval. Statistical analysis was performed using Dunn’s test without correction for multiple comparisons. The *p*-values are indicated as follows: * for 0.2 > *p* > 0.05, ** for 0.05 > *p* ≥ 0.01, *** for 0.01 > *p* ≥ 0.001, and **** for *p* < 0.001.

**Figure 4 toxins-17-00464-f004:**
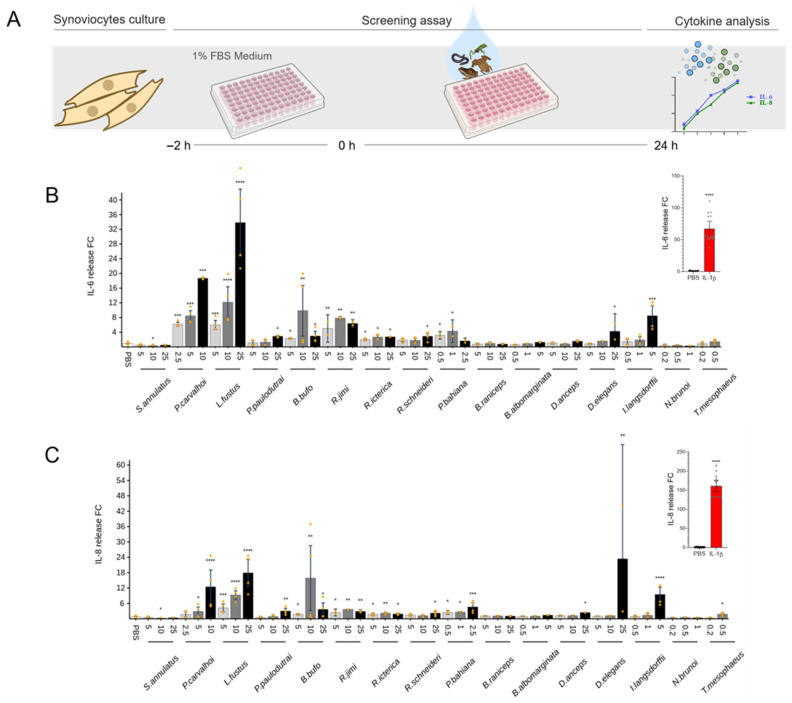
Effect of amphibian secretions on release of IL-6 and IL-8 by synoviocytes. Synoviocyte culture supernatants were collected 24 h after incubation with each amphibian secretion to assess the content of cytokines. (**A**) Assay workflow. (**B**) IL-6. (**C**) IL-8. IL-1β was used as a positive control, and the results are represented in the small graph on the right. Each data point is presented as the mean fold change (FC) from the replicates of two independent experiments relative to cytokine levels of controls (cells in PBS), with error bars representing the confidence interval. Statistical analysis was performed using Dunn’s test without correction for multiple comparisons. The *p*-values are indicated as follows: * for 0.2 > *p* > 0.05, ** for 0.05 > *p* ≥ 0.01, *** for 0.01 > *p* ≥ 0.001, and **** for *p* < 0.001.

**Figure 5 toxins-17-00464-f005:**
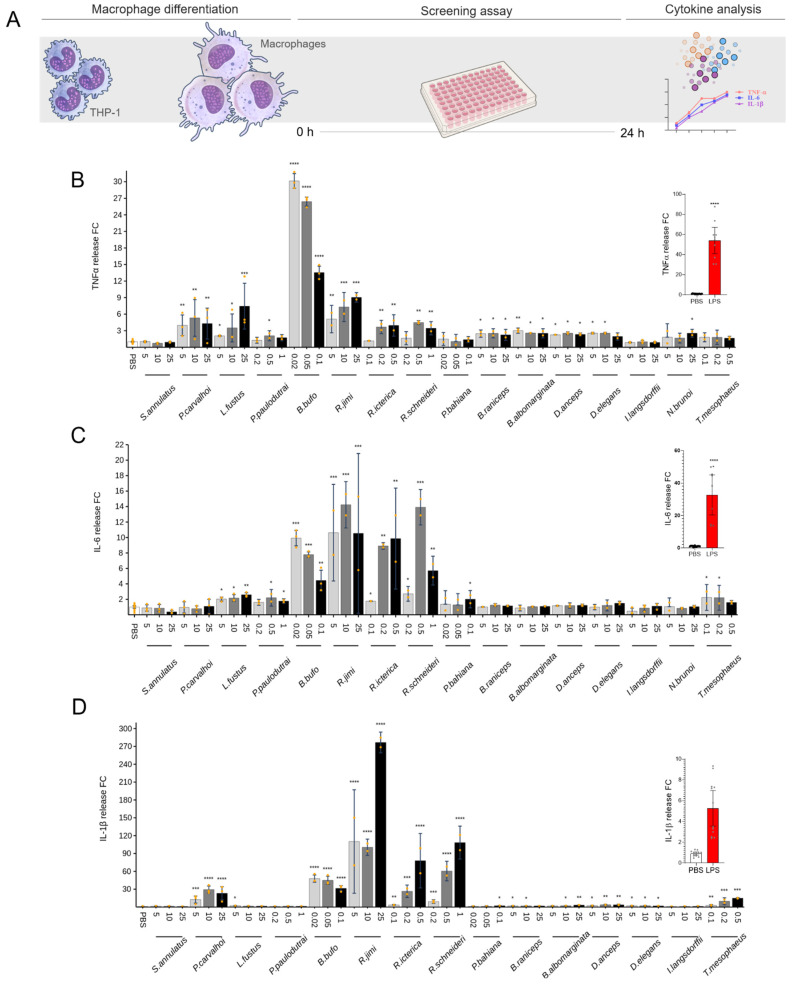
Effect of amphibian secretions on release of TNF-α, IL-6, and IL-1β by macrophages. Macrophage culture supernatants were collected 24 h after incubation with each secretion, to assess the content of the cytokines. (**A**) Assay workflow. (**B**) TNF-α. (**C**) IL-6. (**D**) IL-1β. LPS was used as positive control, and the results are represented in the small graphs on the right. Each data point is presented as the mean fold change (FC) from the replicates of two independent experiments relative to cytokine levels of controls (cells in PBS), with error bars representing the confidence interval. Statistical analysis was performed using Dunn’s test without correction for multiple comparisons. The *p*-values are indicated as follows: * for 0.2 > *p* > 0.05, ** for 0.05 > *p* ≥ 0.01, *** for 0.01 > *p* ≥ 0.001, and **** for *p* < 0.001.

**Figure 6 toxins-17-00464-f006:**
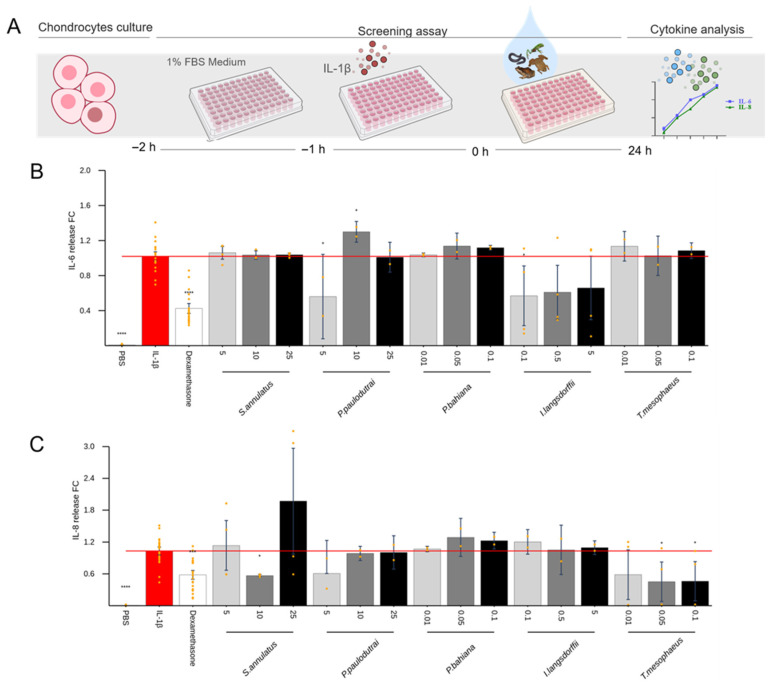
Effect of skin secretions on IL-1β-stimulated chondrocytes. Chondrocytes were treated with IL-1β (1.0 ng/mL) for 1 h. After, dexamethasone (1 µM) or skin secretions were added and incubated for 24 h (**A**). Multiplex evaluated the production of (**B**) IL-6 and (**C**) IL-8 in cell-free supernatants. Each data point is presented as the mean fold change (FC) from the replicates of two independent experiments relative to cytokine levels of positive controls (cells treated with IL-1β), with error bars representing the confidence interval. Statistical analysis was performed using Dunn’s test without correction for multiple comparisons. The *p*-values are indicated as follows: * for 0.2 > *p* > 0.05, *** for 0.01 > *p* ≥ 0.001, and **** for *p* < 0.001.

**Figure 7 toxins-17-00464-f007:**
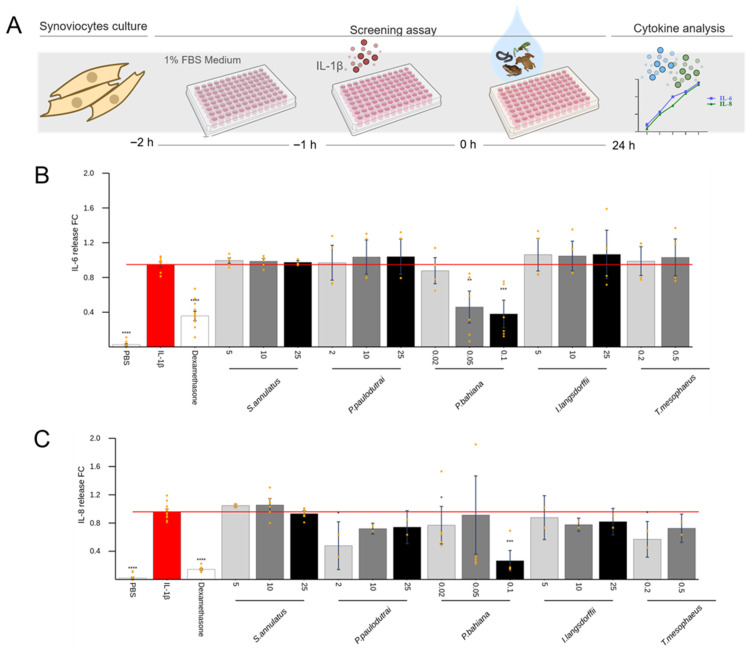
Effect of skin secretions on IL-1β-stimulated synoviocytes. Synoviocytes were treated with IL-1β (1.0 ng/mL) for 1 h. After, dexamethasone (1 µM) or skin secretions were added and incubated for 24 h (**A**). Multiplex evaluated the production of (**B**) IL-6 and (**C**) IL-8 in cell-free supernatants. Each data point is presented as the mean fold change (FC) from the replicates of two independent experiments relative to cytokine levels of positive controls (cells treated with IL-1β), with error bars representing the confidence interval. Statistical analysis was performed using Dunn’s test without correction for multiple comparisons. The *p*-values are indicated as follows: * for 0.2 > *p* > 0.05, ** for 0.05 > *p* ≥ 0.01, *** for 0.01 > *p* ≥ 0.001, and **** for *p* < 0.001.

**Figure 8 toxins-17-00464-f008:**
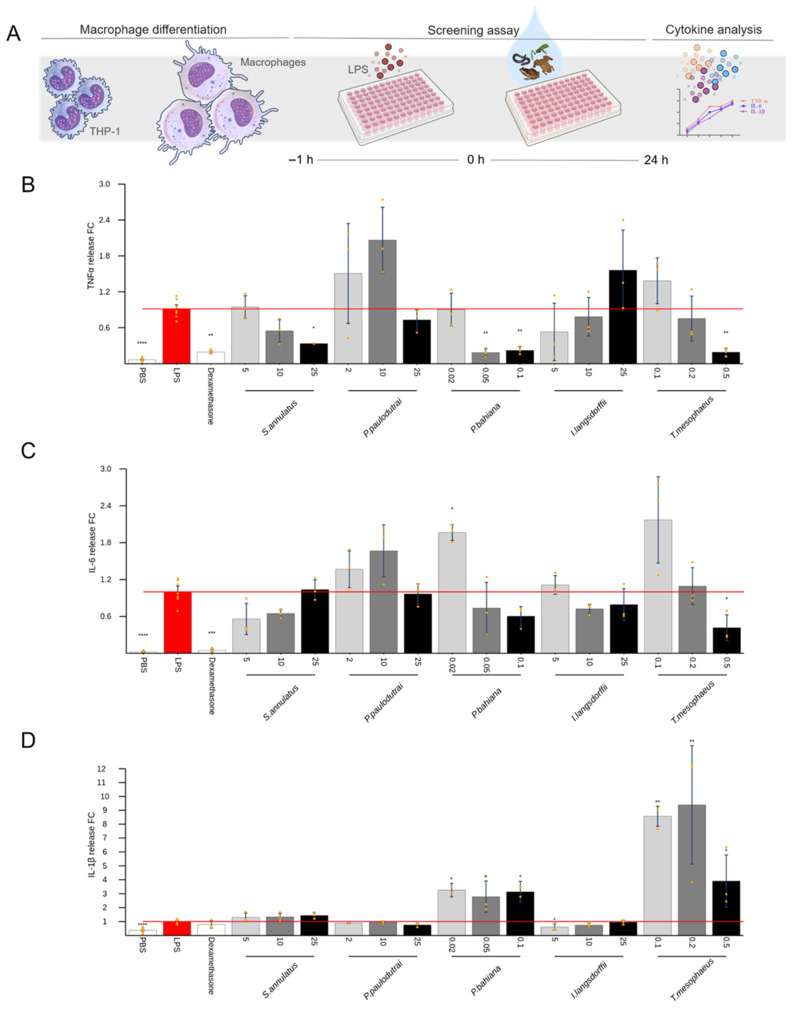
Effect of skin secretions on LPS-stimulated THP-1 macrophages. Macrophages were treated with LPS (10 ng/mL) for 1 h. After, dexamethasone (1 µM) or skin secretions were added and incubated for 24 h (**A**). The production of (**B**) TNF-α, (**C**) IL-6, and (**D**) IL-1β was evaluated by Multiplex in cell-free supernatants. Each data point is presented as the mean fold change (FC) from the replicates of two independent experiments relative to cytokine levels of positive controls (cells treated with LPS), with error bars representing the confidence interval. Statistical analysis was performed using Dunn’s test without correction for multiple comparisons. The *p*-values are indicated as follows: * for 0.20 > *p* ≥ 0.05, ** for 0.05 > *p* ≥ 0.01, *** for 0.01 > *p* ≥ 0.001, and **** for *p* < 0.001.

## Data Availability

The original contributions presented in the study are publicly available. This data can be found at https://doi.org/10.5281/zenodo.16412758.

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
