# Peer review of "Biodiversity-Driven Screening of Amphibian Skin Secretions for Inflammatory Modulation in Joint Diseases"

_toxins, 2025, doi:10.3390/toxins17090464_

Round 1
Reviewer 1 Report
Comments and Suggestions for Authors
The article "Biodiversity-Driven Screening of Amphibian Skin Secretions 3 for Inflammatory Modulation in Joint Diseases" includes a study on the effects of skin secretions from amphibians collected in Brazil for consideration as a source of bioactive compounds.
Regarding the abstract and introduction sections, I have no suggestions to make; I believe they contain the necessary information for the reader to quickly understand what the article is about.
Regarding the results section, the figures included should be distributed among the author-generated sections where they reflect the results related to different individual objectives. Furthermore, in Figure 1, the species can be separated from the number on the horizontal axis to make it more visible and even maintain the same format as Figure 5.
In the discussion section of the text, the figures that support the findings should be included. I don't need to make any suggestions in the content, as the results have been included and compared with other studies.
Some error: Line 524, insert a correct unit .
The conclusions should be expanded, for example, by reflecting the most active genera or species.
Author Response
Dear reviewer, thank you very much for taking the time to review our manuscript. Detailed responses are provided below, and the corresponding revisions and corrections are highlighted in yellow in the resubmitted files.
The article "Biodiversity-Driven Screening of Amphibian Skin Secretions 3 for Inflammatory Modulation in Joint Diseases" includes a study on the effects of skin secretions from amphibians collected in Brazil for consideration as a source of bioactive compounds.
Regarding the abstract and introduction sections, I have no suggestions to make; I believe they contain the necessary information for the reader to quickly understand what the article is about.
Comments 1. Regarding the results section, the figures included should be distributed among the author-generated sections where they reflect the results related to different individual objectives.
Response 1. Thank you for your observation but it was the journal that edited the manuscript and the inclusion of the figures. We changed figure positions as recommended.
Comments 2. Furthermore, in Figure 1, the species can be separated from the number on the horizontal axis to make it more visible and even maintain the same format as Figure 5.
Response 2: The reviewer should be referring to Figure 2 (not Figure 1). The horizontal axis has been changed in the format of figure 3-8.
Comments 3. In the discussion section of the text, the figures that support the findings should be included. I don't need to make any suggestions in the content, as the results have been included and compared with other studies.
Response 3: Thank you for the suggestion, we included the figure references in the discussion:
Comments 4. Some error: Line 524, insert a correct unit .
Response 4: We corrected the unit (uM > µM), thank you
Comments 5. The conclusions should be expanded, for example, by reflecting the most active genera or species.
Response 5: Thank you for your suggestion. The following sentence was included in the Conclusion section:
Of note, Pipa carvalhoi and species from families Leptodactylidae and Bufonidae, exhibited a significant pro-inflammatory effect by inducing cytokine release in the cells. In contrast, others, including Siphonops annulatus, Phyllomedusa bahiana, and Trachycephalus. mesophaeus, were able to reduce cytokine release by activated cells, suggesting a likely anti-inflammatory effect. Interestingly, the latter two skin secretions decreased TNF-α and IL-6 production while simultaneously increasing IL-1β synthesis, indicating the presence of both pro- and anti-inflammatory substances within the secretions.
Reviewer 2 Report
Comments and Suggestions for Authors
Comments to the Author
This manuscript explores the potential of amphibian skin secretions as modulators of inflammatory in joints diseases. This study is well structured, with a comprehensive methodological approach and clear presentation of the findings. This manuscript is of interest in the toxinology field and inflammation research. However, I have several comments and suggestions:
Specific comments:
- The number of individuals of each amphibian specie and biological characteristics of the animals such us sex, age, size are not reported. Voucher specimen deposition is also not mentioned, and some taxonomic identification or geographical origin remain incomplete.
- For secretion collection, the stimulation procedure is described only in general terms, detail on duration, intensity or water volume are not reported.
- The number of biological or technical replicates only is specified in the figure captions. I suggest incorporating this information in the methodology.
- In the Methods section 5.6, cytokine concentrations are described as “presented as fold change compared with negative or positive control”, which is ambiguous. In figure 3, however, it is clearly stated that values are expressed relative to the negative control. I recommended revising the Methods to consistently indicate which control was used as reference in each assay.
- The word “venom” appears in the Cell viability section, although the study is focused on amphibian skin secretions. For accuracy and consistency, I suggest replacing venom with skin secretions in this instance.
- In the Cell viability assay section, the authors define appropriately nontoxic concentrations as those maintaining > 85% viability relative to saline treated controls. However, no positive control for cell death is included. Incorporating a cytotoxic agent would strengthen the assay validation by confirming the sensitivity and dynamic range of the MTT assay.
- Several figure legends include extensive methodological details that would be more appropriate in methods sections.
- The legend of figure 2 does not clarify what the heatmap scale (20-120) represents. This is inconsistent with the Methods sections, where it is stated that viability data were scaled by z-score.
- In the cell culture section, numerical notations lack superscripts (“at a density of 1 × 10⁴ cells/well. I suggest correcting the formatting of exponents throughout the manuscript to maintain consistency and clarity.
- Some parentheses appear unnecessary or inconsistently used, particularly in Figure 1 and in line 441. I recommend revising and standardizing the use of parentheses throughout the manuscript to improve readability.
- The abbreviation for unidentified species is inconsistently written without the period in some instances (Leptodactylus sp). According to taxonomic conventions, it should always be formatted as sp. with a period.
- Although the data are presented as triplicates of two independent experiments, this corresponds to only two biological replicates. Under these conditions, the use of one-way ANOVA with Dunnett`s tets is not statistically robust, as the assumptions of normality and variance homogeneity cannot be reliably assessed. At least three independent experiments are required to justify ANOVA use.
Author Response
Dear reviewer, thank you very much for taking the time to review our manuscript. Detailed responses are provided below, and the corresponding revisions and corrections are highlighted in yellow in the resubmitted files.
This manuscript explores the potential of amphibian skin secretions as modulators of inflammatory in joints diseases. This study is well structured, with a comprehensive methodological approach and clear presentation of the findings. This manuscript is of interest in the toxinology field and inflammation research. However, I have several comments and suggestions:
Specific comments:
Comments 1. The number of individuals of each amphibian specie and biological characteristics of the animals such us sex, age, size are not reported. Voucher specimen deposition is also not mentioned, and some taxonomic identification or geographical origin remain incomplete.
Response 1: Thank you for the observation. All relevant information regarding the animals, secretion collection, and voucher numbers is provided in the supplementary material (Table 1). The number of specimens used for skin secretion extraction ranged from 3 to 6 per species, depending on specimen availability. All individuals were adults, and their sex was not determined.
Comments 2. For secretion collection, the stimulation procedure is described only in general terms, detail on duration, intensity or water volume are not reported.
Response 2: Thank you for the observation: details of secretion collection were included in the MM section as follows:
Briefly, the crude skin secretion of the amphibians was obtained by gentle manual stimulus of the whole body of the animals, one at a time, for 5 minutes inside a plastic bag containing 50 mL of deionized water, forming a pool that was posteriorly freezed at -20°C, stored at -80°C and lyophilized just before the experiments. For Rinella jimi, R. shneideri and R. icterica, the secretion was obtained exclusively from the parotoid macroglands by manual compression and directed collected in plastic tubes. The lyophilized secretion of Bufo bufo was purchased from Latoxan Laboratories S.A.S. (Portes lès Valence, France, product # ID L3101).
Comments 3. The number of biological or technical replicates only is specified in the figure captions. I suggest incorporating this information in the methodology.
Response 3: Thank you for the observation. We included in the MM section the information: All assays were conducted in triplicate in two independent experiments to ensure reproducibility.
Comments 4. In the Methods section 5.6, cytokine concentrations are described as “presented as fold change compared with negative or positive control”, which is ambiguous. In figure 3, however, it is clearly stated that values are expressed relative to the negative control. I recommended revising the Methods to consistently indicate which control was used as reference in each assay.
Response 4: Thank you for your suggestion, we revised the Methods indicating the fold change calculation as follows:
Cytokine concentrations are expressed as fold change (FC). For the analysis of pro-inflammatory effects, FC was calculated by dividing the cytokine concentration in pg/mL obtained for each sample by that of the negative control (cells treated with PBS). For the analysis of anti-inflammatory effects, values from each sample were divided by those of the positive controls, consisting of cells stimulated with IL-1β for chondrocytes and synoviocytes, LPS for macrophages, or glycated collagen for neurons.
Comments 5. The word “venom” appears in the Cell viability section, although the study is focused on amphibian skin secretions. For accuracy and consistency, I suggest replacing venom with skin secretions in this instance.
Response 5: Thank you for the observation. Venom was substituted for amphibian skin secretion in the cell viability section and throughout the text.
Comments 6. In the Cell viability assay section, the authors define appropriately nontoxic concentrations as those maintaining > 85% viability relative to saline treated controls. However, no positive control for cell death is included. Incorporating a cytotoxic agent would strengthen the assay validation by confirming the sensitivity and dynamic range of the MTT assay.
Response 6: Thank you for the question. We have added a graph in the supplementary material showing cell death control using staurosporine at 5 μM in macrophages (Figure S2). Unfortunately, we do not have corresponding data for chondrocytes and synoviocytes. Nevertheless, some skin secretions were highly toxic to these cells, inducing ≥80% cell death.
Comments 7. Several figure legends include extensive methodological details that would be more appropriate in methods sections.
Response 7: Thank you for the suggestion. However, the journals require that figure legends contain all the necessary information for proper understanding. Nonetheless, we have excluded some details from the legends in Figures 6 - 8.
Comments 8. The legend of figure 2 does not clarify what the heatmap scale (20-120) represents. This is inconsistent with the Methods sections, where it is stated that viability data were scaled by z-score.
Response 8: Thank you for the observation. The Methods section and the legend of Figure 2 were corrected as follows:
Methods: Cell viability data are expressed as mean ± standard deviation from two independent experiments. For data visualization, heatmaps were generated in RStudio using the heatmap package, with a color scale ranging from 20% (red) to 120% (green) viability relative to control.
Legend of Figure 2: Heatmap summarizing the cytotoxic effects of the highest concentration of each amphibian skin secretion tested on macrophages, synoviocytes, and chondrocytes. The color scale represents cell viability compared to control, ranging from 20% (red) to 120% (green).
Comments 9. In the cell culture section, numerical notations lack superscripts (“at a density of 1 × 10⁴ cells/well. I suggest correcting the formatting of exponents throughout the manuscript to maintain consistency and clarity.
Response 9: Thank you for the observation. Exponents were corrected.
Comments 10. Some parentheses appear unnecessary or inconsistently used, particularly in Figure 1 and in line 441. I recommend revising and standardizing the use of parentheses throughout the manuscript to improve readability.
Response 10: We appreciate the suggestion and have removed the parentheses from all figures and opened the parenthesis in line 441.
Comments 11. The abbreviation for unidentified species is inconsistently written without the period in some instances (Leptodactylus sp). According to taxonomic conventions, it should always be formatted as sp. with a period.
Response 11: We included the species name Leptodactylus fustus.
Comments 12. Although the data are presented as triplicates of two independent experiments, this corresponds to only two biological replicates. Under these conditions, the use of one-way ANOVA with Dunnett`s tets is not statistically robust, as the assumptions of normality and variance homogeneity cannot be reliably assessed. At least three independent experiments are required to justify ANOVA use.
Response 12: We thank the reviewer for this observation. The non-parametric test applied is described in the Materials and Methods section as follows:
Data are presented as the mean fold change from the triplicates of two independent experiments relative to cytokine, with error bars representing the confidence interval. We applied the two-tailed Student's t-test with a confidence level of 80%. Since many sample sizes were below 5, we used the non-parametric Kruskal-Wallis test followed by Dunn’s test, without duplicate corrections, for a semi-quantitative assessment of the screening. Statistical significance is indicated by the following p-values: * for 0.20 > p ≥ 0.05, ** for 0.05 > p ≥ 0.01, *** for 0.01 > p ≥ 0.001, and **** for p < 0.001.
Reviewer 3 Report
Comments and Suggestions for Authors
The manuscript entitled “Biodiversity-Driven Screening of Amphibian Skin Secretions for Inflammatory Modulation in Joint diseases” (toxins-3808457) reports the cytotoxicity of amphibian skin secretions on human cells involved in joint diseases. It provides new insights into bioactive compounds of amphibians cutaneous secretions with pharmacological applications. However, it needs to be revised before the formal acceptance for publication. Some comments and suggestions are shown in the following:
- For cell viability data, three independent samples are essential.
- Statistical comparisons should be performed in Figure 2 to demonstrate whether the observed differences are statistically significant.
- For the evaluation of cytokines release, each standard curve of cytokines should be provided.
- Where relevant, please add the data points on bar graphs to show the distribution of the data in addition to the error bars.
- The SDS-PAGE image appears to be cropped. Please provide the full, uncropped gel image.
- The article contains some inaccuracies: 1) After first use, genus names may be abbreviated. Additionally, the Latin name of the same genus can be abbreviated. For example, In lines 132 and 142, “Rhinella icterica” should be revised as “R. icterica”. 2) The content in lines 125-126 should be relocated to the Discussion section rather than retained in the Results. 3) For the content in lines 155-157, please provide relevant data. 4) In the text, “p” should be italicized. 5) For consistency in the text, “#” should be revised as “*”. 6) In lines 246, 254, and 262, “1 × 104” should be revised as “1 × 104”. Similar sentences also need to be modified in the text. In a word, the English writing needs further improvement.
- For the references, the format and specification need to be unified following the needs of this journal. Some references are incomplete, including page numbers from start to end.
Author Response
Dear reviewer, thank you very much for taking the time to review our manuscript. Detailed responses are provided below, and the corresponding revisions and corrections are highlighted in yellow in the resubmitted files.
The manuscript entitled “Biodiversity-Driven Screening of Amphibian Skin Secretions for Inflammatory Modulation in Joint diseases” (toxins-3808457) reports the cytotoxicity of amphibian skin secretions on human cells involved in joint diseases. It provides new insights into bioactive compounds of amphibians cutaneous secretions with pharmacological applications. However, it needs to be revised before the formal acceptance for publication. Some comments and suggestions are shown in the following:
Comments 1. For cell viability data, three independent samples are essential.
Response 1: Thank you for the observation. As this is a screening study, an initial test was necessary to identify the most promising skin secretions. We acknowledge that larger sample sizes could further strengthen the analysis; however, given the rarity and limited availability of these secretions, we considered that testing in duplicate at three distinct concentrations, with two independent replicates for each condition, provided a reasonable basis for this preliminary selection process.
Comments 2. Statistical comparisons should be performed in Figure 2 to demonstrate whether the observed differences are statistically significant.
Response 2: we applied a non parametric test that was included in the MM section and in the legend of Figure 2 as follows:
For MM : Data are presented as the mean % viability from the replicates of two independent experiments, with error bars representing the confidence interval. We employed the t-student test with a confidence level of 80%. Given the sample sizes, we used the non-parametric Kruskal-Wallis test followed by Dunn’s test, without duplicate corrections, for a semi-quantitative assessment of the cytotoxicity . To determine statistical significance, refer to the p-values: * for 0.20 > p > 0.05, ** for 0.05 > p > 0.01, *** for 0.01 > p > 0.001], and **** for p-value < 0.001.
For the figure legend: Data are presented as means ± SD of two independent samples. Dunn’s test, without correction for duplicate comparisons, p-values: * for 0.2> p > 0.05, ** for 0.05 > p ≥ 0.01, ** for 0.01 > p ≥ 0.001, and *** for p < 0.001.
Comments 3. For the evaluation of cytokines release, each standard curve of cytokines should be provided.
Response 3: Representative standard curves of cytokines generated by the Milliplex® multiplex cytokine assay report (#HCYTOMAG-60K) are presented in the Supplementary Material Figure S3.
Comments 4. Where relevant, please add the data points on bar graphs to show the distribution of the data in addition to the error bars.
Response 4: Thank you for your suggestion. We added data points on bar graphs in Figures 2 to 8.
Comments 5. The SDS-PAGE image appears to be cropped. Please provide the full, uncropped gel image.
Response 5: We have sent the uncropped cell image to the journal before and now we included it in Supplementary Material, Figure S1.
Comments 6.1. The article contains some inaccuracies:
1) After first use, genus names may be abbreviated. Additionally, the Latin name of the same genus can be abbreviated. For example, In lines 132 and 142, “Rhinella icterica” should be revised as “R. icterica”. .
Response 6.1: We appreciate the reviewer’s observation. In this new version, we confirm that we have adhered to the guidelines set forth by the animal biology journals in which we regularly publish, including recent work in Toxins. We have followed the protocol of writing out the full genus and species names the first time they appear in each paragraph. For all subsequent mentions within the same paragraph, we use the abbreviation for the genus. This guideline has been consistently applied throughout the text.
Comments 6.2. The content in lines 125-126 should be relocated to the Discussion section rather than retained in the Results.
Response 6.2: We agree that this information may be of interest; however, as it only refers to the absence of these secretions in subsequent trials, we believe it is not essential to include it in the Discussion.
Comments 6.3. For the content in lines 155-157, please provide relevant data.
Response 6.3: The determination of non cytotoxic doses of secretions are presented in Figure S4 in Supplementary material.
Comments 6.4. In the text, “p” should be italicized.
Response 6.4: Thank you, p was italicized in the Figure legends and in MM
Comments 6.5. For consistency in the text, “#” should be revised as “*”.
Response 6.5: # was changed to * in Figures 6 to 8.
Comments 6.6. In lines 246, 254, and 262, “1 × 104” should be revised as “1 × 104”.
Response 6.6: Thank you, the exponents were corrected
Similar sentences also need to be modified in the text.
In a word, the English writing needs further improvement.
The manuscript has been reviewed by native English-speaking co-authors and further revised using the Grammarly app.
Comments 7. For the references, the format and specification need to be unified following the needs of this journal. Some references are incomplete, including page numbers from start to end.
Response 7: We appreciate your observation and double checked the references for accuracy. However, we used the Zotero tool and retrieved the papers by DOI, following the specific reference style required by Toxins. If the number of pages was not pulled automatically, it is because this information doesn’t exist.
Round 2
Reviewer 3 Report
Comments and Suggestions for Authors
In lines 183, 216, 226, 238, 267 and 292, "* for 0.20 > p≥ 0.05, ** for 0.05 > p≥ 0.01, ** for 0.01 > p≥ 0.001, and *** for p< 0.001" should be revised as "* for 0.20 > p ≥ 0.05, ** for 0.05 > p ≥ 0.01, *** for 0.01 > p ≥ 0.001, and **** for p < 0.001" (line 638).
Author Response
Comment 1: In lines 183, 216, 226, 238, 267 and 292, "* for 0.20 > p≥ 0.05, ** for 0.05 > p≥ 0.01, ** for 0.01 > p≥ 0.001, and *** for p< 0.001" should be revised as "* for 0.20 > p ≥ 0.05, ** for 0.05 > p ≥ 0.01, *** for 0.01 > p ≥ 0.001, and **** for p < 0.001" (line 638).
Response 1: Thank you for the observation. We corrected lines 183, 216, 226, 238, 267, and 292 as suggested, and highlighted the changes in yellow.